# Variational Planning for Graph-based MDPs

**Qiang Cheng**[†]        **Qiang Liu**[‡]        **Feng Chen**[†]        **Alexander Ihler**[‡]

[†]Department of Automation, Tsinghua University
[‡]Department of Computer Science, University of California, Irvine
[†]{cheng-q09@mails., chenfeng@mail.}tsinghua.edu.cn
[‡]{qliu1@,ihler@ics.}uci.edu

## Abstract

Markov Decision Processes (MDPs) are extremely useful for modeling and solving sequential decision making problems. Graph-based MDPs provide a compact representation for MDPs with large numbers of random variables. However, the complexity of exactly solving a graph-based MDP usually grows exponentially in the number of variables, which limits their application. We present a new variational framework to describe and solve the planning problem of MDPs, and derive both exact and approximate planning algorithms. In particular, by exploiting the graph structure of graph-based MDPs, we propose a factored variational value iteration algorithm in which the value function is first approximated by the multiplication of local-scope value functions, then solved by minimizing a Kullback-Leibler (KL) divergence. The KL divergence is optimized using the belief propagation algorithm, with complexity exponential in only the cluster size of the graph. Experimental comparison on different models shows that our algorithm outperforms existing approximation algorithms at finding good policies.

## 1   Introduction

Markov Decision Processes (MDPs) have been widely used to model and solve sequential decision making problems under uncertainty, in fields including artificial intelligence, control, finance and management (Puterman, 2009, Barber, 2011). However, standard MDPs are described by explicitly enumerating all possible states of variables, and are thus not well suited to solve large problems. Graph-based MDPs (Guestrin et al., 2003, Forsell and Sabbadin, 2006) provide a compact representation for large and structured MDPs, where the transition model is explicitly represented by a dynamic Bayesian network. In graph-based MDPs, the state is described by a collection of random variables, and the transition and reward functions are represented by a set of smaller (local-scope) functions. This is particularly useful for spatial systems or networks with many "local" decisions, each affecting small sub-systems that are coupled together and interdependent (Nath and Domingos, 2010, Sabbadin et al., 2012).

The graph-based MDP representation gives a compact way to describe a structured MDP, but the complexity of exactly solving such MDPs typically still grows exponentially in the number of state variables. Consequently, graph-based MDPs are often approximately solved by enforcing context-specific independence or function-specific independence constraints (Sigaud et al., 2010). To take advantage of context-specific independence, a graph-based MDP can be represented using decision trees or algebraic decision diagrams (Bahar et al., 1993), and then solved by applying structured value iteration (Hoey et al., 1999) or structured policy iteration (Boutilier et al., 2000). However, in the worst case, the size of the diagram still increases exponentially with the number of variables. Alternatively, methods based on function-specific independence approximate the value function by a linear combination of basis functions (Koller and Parr, 2000, Guestrin et al., 2003). Exploiting function-specific independence, a graph-based MDP can be solved using approximate linear programming (Guestrin et al., 2003, 2001, Forsell and Sabbadin, 2006), approximate policy itera-

tion (Sabbadin et al., 2012, Peyrard and Sabbadin, 2006) and approximate value iteration (Guestrin et al., 2003). Among these, the approximate linear programming algorithm in Guestrin et al. (2003, 2001) has an exponential number of constraints (in the treewidth), and thus cannot be applied to general MDPs with many variables. The approximate policy iteration algorithm in Sabbadin et al. (2012), Peyrard and Sabbadin (2006) exploits a mean field approximation to compute and update the local policies; unfortunately this can give loose approximations.

In this paper, we propose a variational framework for the MDP planning problem. This framework provides a new perspective to describe and solve graph-based MDPs where both the state and decision spaces are structured. We first derive a variational value iteration algorithm as an exact planning algorithm, which is equivalent to the classical value iteration algorithm. We then design an approximate version of this algorithm by taking advantage of the factored representation of the reward and transition functions, and propose a factored variational value iteration algorithm. This algorithm treats the value function as a unnormalized distribution and approximates it using a product of local-scope value functions. At each step, this algorithm computes the value function by minimizing a Kullback-Leibler divergence, which can be done using a belief propagation algorithm for influence diagram problems (Liu and Ihler, 2012) . In comparison with the approximate linear programming algorithm (Guestrin et al., 2003) and the approximate policy iteration algorithm (Sabbadin et al., 2012) on various graph-based MDPs, we show that our factored variational value iteration algorithm generates better policies.

The remainder of this paper is organized as follows. The background and some notation for graph-based MDPs are introduced in Section 2. Section 3 describes a variational view of planning for finite horizon MDPs, followed by a framework for infinite MDPs in Section 4. In Section 5, we derive an approximate algorithm for solving infinite MDPs based on the variational perspective. We show experiments to demonstrate the effectiveness of our algorithm in Section 6.

## 2 Markov Decision Processes and Graph-based MDPs

### 2.1 Markov Decision Processes

A Markov Decision Process (MDP) is a discrete time stochastic control process, where the system chooses the decisions at each step to maximize the overall reward. An MDP can be characterized by a four tuple $(\mathcal{X}, \mathcal{D}, R, T)$, where $\mathcal{X}$ represents the set of all possible states; $\mathcal{D}$ is the set of all possible decisions; $R : \mathcal{X} \times \mathcal{D} \to \mathbb{R}$ is the reward function of the system, and $R(\mathbf{x}, \mathbf{d})$ is the reward of the system after choosing decision $\mathbf{d}$ in state $\mathbf{x}$; $T : \mathcal{X} \times \mathcal{D} \times \mathcal{X} \to [0, 1]$ is the transition function, and $T(\mathbf{y}|\mathbf{x}, \mathbf{d})$ is the probability that the system arrives at state $\mathbf{y}$, given that it starts from $\mathbf{x}$ upon executing decision $\mathbf{d}$. A *policy* of the system is a mapping from the states to the decisions $\pi(\mathbf{x}) : \mathcal{X} \to \mathcal{D}$ so that $\pi(\mathbf{x})$ tells the decision chosen by the system in state $\mathbf{x}$. The graphical representation of an MDP is shown in Figure 1(a).

We consider the case of an MDP with infinite horizon, in which the future rewards are discounted exponentially with a discount factor $\gamma \in [0, 1]$. The task of the MDP is to choose the best stationary policy $\pi^*(\mathbf{x})$ that maximizes the expected discounted reward on the infinite horizon. The value function $v^*(\mathbf{x})$ of the best policy $\pi^*(\mathbf{x})$ then satisfies the following Bellman equation:

$$v^*(\mathbf{x}) = \max_{\pi(\mathbf{x})} \sum_{\mathbf{y} \in \mathcal{X}} T(\mathbf{y}|\mathbf{x}, \pi(\mathbf{x})) \left(R(\mathbf{x}, \pi(\mathbf{x})) + \gamma v^*(\mathbf{y})\right), \quad (1)$$

where $v^*(\mathbf{x}) = v^*(\mathbf{y}), \forall \mathbf{x} = \mathbf{y}$. The Bellman equation can be solved using stochastic dynamic programming algorithms such as value iteration and policy iteration, or linear programming algorithms (Puterman, 2009).

### 2.2 Graph-based MDPs

We assume that the full state $\mathbf{x}$ can be represented as a collection of state variables $x_i$, so that $\mathcal{X}$ is a Cartesian product of the domains of the $x_i$: $\mathcal{X} = \mathcal{X}_1 \times \mathcal{X}_2 \times \cdots \times \mathcal{X}_N$, and similarly for $\mathbf{d}$: $\mathcal{D} = \mathcal{D}_1 \times \mathcal{D}_2 \times \cdots \times \mathcal{D}_N$. We consider the following particular factored form for MDPs: for each variable $i$, there exist neighborhood sets $\Gamma_i$ (including $i$) such that the value of $x_i^{t+1}$ depends only on the variable $i$'s neighborhood, $\mathbf{x}^t[\Gamma_i]$, and the $i$th decision $d_i^t$. Then, we can write the transition function in a factored form:

$$T(\mathbf{y}|\mathbf{x}, \mathbf{d}) = \prod_{i=1}^{N} T_i(y_i|\mathbf{x}[\Gamma_i], d_i), \quad (2)$$

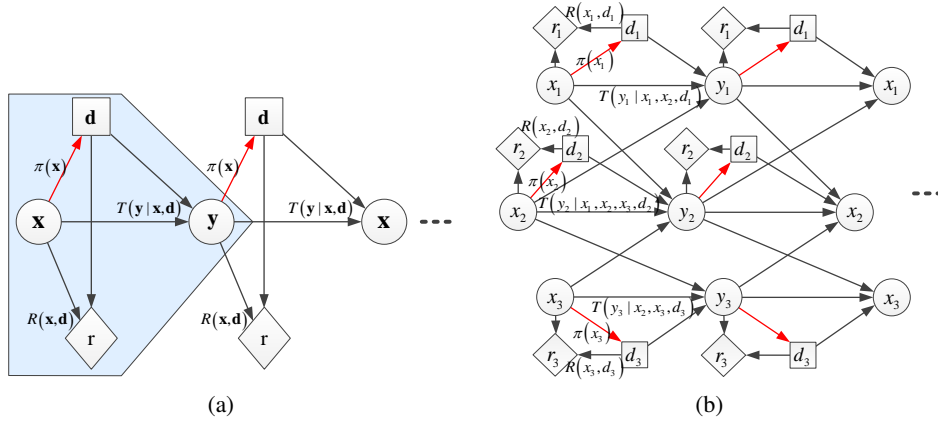

Figure 1: (a) A Markov decision process; (b) A graph-based Markov decision process.

where each factor is a local-scope function $T_i : \mathcal{X}\left[\Gamma_i\right] \times \mathcal{D}_i \times \mathcal{X}_i \rightarrow [0,1], \forall i \in \{1, 2, \ldots, N\}$. We also assume that the reward function is the sum of $N$ local-scope rewards:

$$R\left(\mathbf{x}, \mathbf{d}\right) = \sum_{i=1}^{N} R_i\left(x_i, d_i\right), \tag{3}$$

with local-scope functions $R_i : \mathcal{X}_i \times \mathcal{D}_i \rightarrow \mathbb{R}, \forall i \in \{1, 2, \ldots, N\}$.

To summarize, a graph-based Markov decision process is characterized by the following parameters: $(\{\mathcal{X}_i : 1 \leq i \leq N\}; \{\mathcal{D}_i : 1 \leq i \leq N\}; \{R_i : 1 \leq i \leq N\}; \{\Gamma_i : 1 \leq i \leq N\}; \{T_i : 1 \leq i \leq N\})$. Figure 1(b) gives an example of a graph-based MDP. These assumptions for graph-based MDPs can be easily generalized, for example to include $T_i$ and $R_i$ that depend on arbitrary sets of variables and decisions, using some additional notation.

The optimal policy $\pi\left(\mathbf{x}\right)$ cannot be explicitly represented for large graph-based MDPs, since the number of states grows exponentially with the number of variables. To reduce complexity, we consider a particular class of local policies: a policy $\pi\left(\mathbf{x}\right) : \mathcal{X} \rightarrow \mathcal{D}$ is said to be local if decision $d_i$ is made using only the neighborhood $\Gamma_i$, so that $\pi\left(\mathbf{x}\right) = \left(\pi_1\left(\mathbf{x}\left[\Gamma_1\right]\right), \pi_2\left(\mathbf{x}\left[\Gamma_2\right]\right), \ldots, \pi_N\left(\mathbf{x}\left[\Gamma_N\right]\right)\right)$ where $\pi_i\left(\mathbf{x}\left[\Gamma_i\right]\right) : \mathcal{X}\left[\Gamma_i\right] \rightarrow \mathcal{D}_i$. The main advantage of local policies is that they can be concisely expressed when the neighborhood sizes $|\Gamma_i|$ are small.

## 3 Variational Planning for Finite Horizon MDPs

In this section, we introduce a variational planning viewpoint of finite MDPs. A finite MDP can be viewed as an influence diagram; we can then directly relate planning to the variational decision-making framework of Liu and Ihler (2012).

Influence diagrams (Shachter, 2007) make use of Bayesian networks to represent structured decision problems under uncertainty. The shaded part in Figure 1(a) shows a simple example influence diagram, with random variables $\{\mathbf{x}, \mathbf{y}\}$, decision variable $\mathbf{d}$ and reward functions $\{R\left(\mathbf{x}, \mathbf{d}\right), v\left(\mathbf{y}\right)\}$. The goal is then to choose a policy that maximizes the expected reward.

The best policy $\pi^t\left(\mathbf{x}\right)$ for a finite MDP can be computed using backward induction (Barber, 2011):

$$v^{t-1}\left(\mathbf{x}\right) = \max_{\pi\left(\mathbf{x}\right)} \sum_{\mathbf{y} \in \mathcal{X}} T\left(\mathbf{y}|\mathbf{x}, \pi\left(\mathbf{x}\right)\right)\left(R\left(\mathbf{x}, \pi\left(\mathbf{x}\right)\right) + \gamma v^t\left(\mathbf{y}\right)\right), \tag{4}$$

Let $p^t\left(\mathbf{x}, \mathbf{y}, \mathbf{d}\right) = T\left(\mathbf{y}|\mathbf{x}, \pi\left(\mathbf{x}\right)\right)\left(R\left(\mathbf{x}, \pi\left(\mathbf{x}\right)\right) + \gamma v^t\left(\mathbf{y}\right)\right)$ be an augmented distribution (see, e.g., Liu and Ihler (2012)). Applying a variational framework for influence diagrams (Liu and Ihler, 2012, Theorem 3.1), the optimal policy can be equivalently solved from the dual form of Eq. (4):

$$\Phi\left(\boldsymbol{\theta}^t\right) = \max_{\boldsymbol{\tau} \in \mathbb{M}} \left\{\left\langle \boldsymbol{\theta}^{\Delta;t}, \boldsymbol{\tau} \right\rangle + H\left(\mathbf{x}, \mathbf{y}, \mathbf{d}; \boldsymbol{\tau}\right) - H\left(\mathbf{d}|\mathbf{x}; \boldsymbol{\tau}\right)\right\}, \tag{5}$$

where $\boldsymbol{\theta}^{\Delta;t}\left(\mathbf{x}, \mathbf{y}, \mathbf{d}\right) = \log p^t\left(\mathbf{x}, \mathbf{y}, \mathbf{d}\right) = \log T\left(\mathbf{y}|\mathbf{x}, \mathbf{d}\right) + \log\left(R\left(\mathbf{x}, \mathbf{d}\right) + \gamma v^t\left(\mathbf{y}\right)\right)$, and $\boldsymbol{\tau}$ is a vector of moments in the *marginal polytope* $\mathbb{M}$ (Wainwright and Jordan, 2008). In a mild abuse of notation, we will use $\boldsymbol{\tau}$ to refer both to the vector of moments and to the maximum entropy

distribution $\boldsymbol{\tau}(\mathbf{x}, \mathbf{y}, \mathbf{d})$ consistent with those moments; $H(\cdot; \boldsymbol{\tau})$ refers to the entropy or conditional entropy of this distribution. See also Wainwright and Jordan (2008), Liu and Ihler (2012) for details.

Let $\boldsymbol{\tau}^t(\mathbf{x}, \mathbf{y}, \mathbf{d})$ be the optimal solution of Eq. (5); then from Liu and Ihler (2012), the optimal policy $\pi^t(\mathbf{x})$ is simply $\arg\max_{\mathbf{d}} \boldsymbol{\tau}^t(\mathbf{d}|\mathbf{x})$. Moreover, the optimal value function $v^{t-1}(\mathbf{x})$ can be obtained from Eq. (5). This result is summarized in the following lemma.

**Lemma 1.** *For finite MDPs with non-stationary policy, the best policy $\pi^t(\mathbf{x})$ and the value function $v^{t-1}(\mathbf{x})$ can be obtained by solving Eq. (5). Let $\boldsymbol{\tau}^t(\mathbf{x}, \mathbf{y}, \mathbf{d})$ be the optimal solution of Eq. (5).*

*(a) The optimal policy can be obtained from $\boldsymbol{\tau}^t(\mathbf{x}, \mathbf{y}, \mathbf{d})$, as $\pi^t(\mathbf{x}) = \arg\max_{\mathbf{d}} \boldsymbol{\tau}^t(\mathbf{d}|\mathbf{x})$.*

*(b) The value function w.r.t. $\pi^t(\mathbf{x})$ can be obtained as $v^{t-1}(\mathbf{x}) = \exp(\Phi(\boldsymbol{\theta}^t)) \boldsymbol{\tau}^t(\mathbf{x})$.*

*Proof.* (a) follows directly from Theorem 3.1 of Liu and Ihler (2012). (b) Note that $T(\mathbf{y}|\mathbf{x}, \pi^t(\mathbf{x}))(R(\mathbf{x}, \pi^t(\mathbf{x})) + \gamma v^t(\mathbf{y})) = \exp(\Phi(\boldsymbol{\theta}^t)) \boldsymbol{\tau}^t(\mathbf{x}, \mathbf{y}, \mathbf{d})$. Making use of Eq. (4), summing over $\mathbf{y}$ and maximizing over $\mathbf{d}$ on $\exp(\Phi(\boldsymbol{\theta}^t)) \boldsymbol{\tau}^t(\mathbf{x}, \mathbf{y}, \mathbf{d})$, we obtain $v^{t-1}(\mathbf{x}) = \exp(\Phi(\boldsymbol{\theta}^t)) \boldsymbol{\tau}^t(\mathbf{x})$. $\square$

## 4 Variational Planning for Infinite Horizon MDPs

Given the variational form of finite MDPs, we now construct a variational framework for infinite MDPs. Compared to the primal form (i.e., Eq. (4)) of finite MDPs, the Bellman equation of an infinite MDP, Eq. (1), has the additional constraint that $v^{t-1}(\mathbf{x}) = v^t(\mathbf{y})$ when $\mathbf{x} = \mathbf{y}$. For an infinite MDP, we can simply consider a two-stage finite MDP with the variational form in Eq. (5), but with this additional constraint. The main result is given by the following theorem.

**Theorem 2.** *Assume $\boldsymbol{\tau}$ and $\Phi$ are the solution of the following optimization problem,*

$$\max_{\boldsymbol{\tau} \in \mathbb{M}, \Phi \in \mathbb{R}} \Phi, \quad subject\ to \quad \Phi = \left\langle \boldsymbol{\theta}^\Delta, \boldsymbol{\tau} \right\rangle + H(\mathbf{x}, \mathbf{y}, \mathbf{d}; \boldsymbol{\tau}) - H(\mathbf{d}|\mathbf{x}; \boldsymbol{\tau}), \quad (6)$$

$$\boldsymbol{\theta}^\Delta = \log T(\mathbf{y}|\mathbf{x}, \mathbf{d}) + \log(R(\mathbf{x}, \mathbf{d}) + \gamma \exp(\Phi) \boldsymbol{\tau}_{\mathbf{x}}(\mathbf{y})), \quad (7)$$

*where $\boldsymbol{\tau}_{\mathbf{x}}$ denotes the marginal distribution on $\mathbf{x}$. With $\boldsymbol{\tau}^*$ being the optimal solution, we have*

*(a) The optimal policy of the infinite MDP can be decoded as $\pi^*(\mathbf{x}) = \arg\max_{\mathbf{d}} \boldsymbol{\tau}^*(\mathbf{d}|\mathbf{x})$.*

*(b) The value function w.r.t. $\pi^*(\mathbf{x})$ is $v^*(\mathbf{x}) = \exp(\Phi) \boldsymbol{\tau}^*(\mathbf{x})$.*

*Proof.* The Bellman equation is equivalent to the backward induction in Eq. (4), subject to an extra constraint that $v^t = v^{t-1}$. The result follows by replacing Eq. (4) with its variational dual (5). $\square$

Like the Bellman equation (4), its dual form (6) also has no closed-form solution. Analogously to the value iteration algorithm for the Bellman equation, Eq. (6) can be solved by alternately fixing $\boldsymbol{\tau}_{\mathbf{x}}(\mathbf{x})$, $\Phi$ in $\boldsymbol{\theta}^\Delta$ and solving Eq. (6) with only the first constraint using some convex optimization technique. However, each step of solving for $\boldsymbol{\tau}$ and $\Phi$ is equivalent to one step of value iteration; if $\boldsymbol{\tau}(\mathbf{x}, \mathbf{y}, \mathbf{d})$ is represented explicitly, it seems to offer no advantage over simply applying the elimination operators as in (4). The usefulness of this form is mainly in opening the door to design new approximations.

## 5 Approximate Variational Algorithms for Graph-based MDPs

The framework in the previous section gives a new perspective on the MDP planning problem, but does not by itself simplify the problem or provide new solution methods. For graph-based MDPs, the sizes of the full state and decision spaces are exponential in the number of variables. Thus, the complexity of exact algorithms is exponentially large. In this section, we present an approximate algorithm for solving Eq. (6), by exploiting the factorization structure of the transition function (2), the reward function (3) and the value function $v(\mathbf{x})$.

Standard variational approximations take advantage of the multiplicative factorization of a distribution to define their approximations. While our (unnormalized) distribution $p(\mathbf{x}, \mathbf{y}, \mathbf{d}) = \exp[\boldsymbol{\theta}^\Delta(\mathbf{x}, \mathbf{y}, \mathbf{d})]$ is structured, some of its important structure comes from additive factors, such as the local-scope reward functions $R_i(x_i, d_i)$ in Eq. (3), and the discounted value function $\gamma v(\mathbf{x})$ in Eq. (1). Computing the sum of these additive factors directly would create a large factor over an unmanageably large variable domain, and destroy most of the useful structure of $p(\mathbf{x}, \mathbf{y}, \mathbf{d})$.

To avoid this effect, we convert the presence of additive factors into multiplicative factors by augmenting the model with a latent "selector" variable, which is similar to that used for the "complete likelihood" in mixture models (Liu and Ihler, 2012). For example, consider the sum of two factors:

$$f(\mathbf{x}) = f_{12}(x_1, x_2) + f_{23}(x_2, x_3) = \sum_{a \in \{0,1\}} (f_{12})^a \cdot (f_{23})^{(1-a)} = \sum_{a \in \{0,1\}} \bar{f}_{12}(a, x_1, x_2) \cdot \bar{f}_{23}(a, x_2, x_3).$$

Introducing the auxilliary variable $a$ converts $f$ into a product of factors, where marginalizing over $a$ yields the original function $f$.

Using this augmenting approach, the additive elements of the graph-based MDP are converted to multiplicative factors, that is $R_i(x_i, d_i) \to \tilde{R}_i(x_i, d_i, a)$, and $\gamma v(\mathbf{x}) \to \tilde{v}_\gamma(\mathbf{x}, a)$. In this way, the parameter $\boldsymbol{\theta}^\Delta$ of a graph-based MDP can be represented as

$$\boldsymbol{\theta}^\Delta(\mathbf{x}, \mathbf{y}, \mathbf{d}, a) = \sum_{i=1}^N \log T_i(y_i | \mathbf{x}[\Gamma_i], d_i) + \sum_{i=1}^N \log \tilde{R}_i(x_i, d_i, a) + \log \tilde{v}_\gamma(\mathbf{y}, a).$$

Now, $p(\mathbf{x}, \mathbf{y}, \mathbf{d}, a) = \exp[\boldsymbol{\theta}^\Delta(\mathbf{x}, \mathbf{y}, \mathbf{d}, a)]$ has a representation in terms of a product of factors. Let

$$\boldsymbol{\theta}(\mathbf{x}, \mathbf{y}, \mathbf{d}, a) = \sum_{i=1}^N \log T_i(y_i | \mathbf{x}[\Gamma_i], d_i) + \sum_{i=1}^N \log \tilde{R}_i(x_i, d_i, a).$$

Before designing the algorithms, we first construct a cluster graph $(\mathcal{G}; \mathcal{C}; \mathcal{S})$ for the distribution $\exp[\boldsymbol{\theta}(\mathbf{x}, \mathbf{y}, \mathbf{d}, a)]$, where $\mathcal{C}$ denotes the set of clusters and $\mathcal{S}$ is the set of separators. (See Liu and Ihler (2012, 2011), Wainwright and Jordan (2008) for more details on cluster graphs.) We assign each decision node $d_i$ to one cluster that contains $d_i$ and its parents $pa(i)$; clusters so assigned are called *decision clusters* $\mathcal{A}$, while other clusters are called *normal clusters* $\mathcal{R}$, so that $\mathcal{C} = \{\mathcal{R}, \mathcal{A}\}$.

Using the structure of the cluster graph, $\boldsymbol{\theta}$ can be decomposed into

$$\boldsymbol{\theta}(\mathbf{x}, \mathbf{y}, \mathbf{d}, a) = \sum_{k \in \mathcal{C}} \boldsymbol{\theta}_{c_k}(\mathbf{x}_{c_k}, \mathbf{y}_{c_k}, \mathbf{d}_{c_k}, a), \tag{8}$$

and the distribution $\boldsymbol{\tau}$ is approximated as

$$\boldsymbol{\tau}(\mathbf{x}, \mathbf{y}, \mathbf{d}, a) = \frac{\prod_{k \in \mathcal{C}} \boldsymbol{\tau}_{c_k}(\mathbf{z}_{c_k})}{\prod_{(kl) \in \mathcal{S}} \boldsymbol{\tau}_{s_{kl}}(\mathbf{z}_{s_{kl}})}, \tag{9}$$

where $\mathbf{z}_{c_k} = \{\mathbf{x}_{c_k}, \mathbf{y}_{c_k}, \mathbf{d}_{c_k}, a\}$. Therefore, instead of optimizing the full distribution $\boldsymbol{\tau}$, we can optimize the collection of marginal distributions $\boldsymbol{\tau} = \{\boldsymbol{\tau}_{c_k}, \boldsymbol{\tau}_{s_k}\}$, with far lower computational cost. These marginals should belong to the local consistency polytope $\mathbb{L}$, which enforces that marginals are consistent on their overlapping sets of variables (Wainwright and Jordan, 2008).

We now construct a reduced cluster graph over $\mathbf{x}$ from the full cluster graph, to serve as the approximating structure of the marginal $\boldsymbol{\tau}(\mathbf{x})$. We assume a factored representation for $\boldsymbol{\tau}(\mathbf{x})$:

$$\boldsymbol{\tau}(\mathbf{x}) = \frac{\prod_{k \in \mathcal{C}} \boldsymbol{\tau}_{c_k}(\mathbf{x}_{c_k})}{\prod_{(kl) \in \mathcal{S}} \boldsymbol{\tau}_{s_{kl}}(\mathbf{x}_{s_{kl}})}, \tag{10}$$

where the $\boldsymbol{\tau}_{c_k}(\mathbf{x}_{c_k})$ is the marginal distribution of $\boldsymbol{\tau}_{c_k}(\mathbf{z}_{c_k})$ on $\mathbf{x}_{c_k}$. Note that Eq. (10) also dictates a factored approximation of the value function $v(\mathbf{x})$, because $v(\mathbf{x}) \approx \exp(\Phi)\boldsymbol{\tau}(\mathbf{x})$. Assume $v_\gamma(\mathbf{x})$ factors into $v_\gamma(\mathbf{x}) = \prod_k v_{c_k}(\mathbf{x}_{c_k})$. Then, the constraint (7) reduces to a set of simpler constraints on the cliques of the cluster graph,

$$\boldsymbol{\theta}_{c_k}^\Delta(\mathbf{x}_{c_k}, \mathbf{y}_{c_k}, \mathbf{d}_{c_k}, a) = \boldsymbol{\theta}_{c_k}(\mathbf{x}_{c_k}, \mathbf{y}_{c_k}, \mathbf{d}_{c_k}, a) + \log v_{c_k, \mathbf{x}}(\mathbf{y}_{c_k}, a), \quad k \in \mathcal{C}. \tag{11}$$

Correspondingly, the constraint (6) can be approximated by

$$\Phi = \sum_{k \in \mathcal{C}} \langle \boldsymbol{\theta}_{c_k}^\Delta, \boldsymbol{\tau}_{c_k} \rangle + \sum_{k \in \mathcal{R}} H_{c_k} + \sum_{k \in \mathcal{D}} H'_{c_k} - \sum_{(kl) \in \mathcal{S}} H_{s_{kl}}, \tag{12}$$

where $H_{c_k}$ is the entropy of variables in cluster $c_k$, $H_{c_k} = H(\mathbf{x}_{c_k}, \mathbf{y}_{c_k}, \mathbf{d}_{c_k}, a; \boldsymbol{\tau})$ and $H'_{c_k} = H(\mathbf{x}_{c_k}, \mathbf{y}_{c_k}, \mathbf{d}_{c_k}, a; \boldsymbol{\tau}) - H(\mathbf{d}_{c_k} | \mathbf{x}_{c_k}; \boldsymbol{\tau})$. With these approximations, we solve the optimization in Theorem 2 using "mixed" belief propagation (Liu and Ihler, 2012) for fixed $\{\boldsymbol{\theta}_{c_k}^\Delta\}$; we then update $\{\boldsymbol{\theta}_{c_k}^\Delta\}$ using the fixed point condition (11). This gives the double loop algorithm in Algorithm 1.

**Algorithm 1** Factored Variational Value Iteration Algorithm

---

**Input:** A graph-based MDP with $(\{\mathcal{X}_i\}; \{\mathcal{D}_i\}; \{R_i\}; \{\Gamma_i\}; \{T_i\})$, the cluster graph $(\mathcal{G}; \mathcal{C}; \mathcal{S})$, and the initial $\{\boldsymbol{\tau}_{c_k}^{t=0}(\mathbf{x}_{c_k}), \forall c_k \in \mathcal{C}\}$.
  Iterate until convergence (for both the outer loop and the inner loop).
1:   Outer loop: Update $\boldsymbol{\theta}_{c_k}^{\Delta;t}$ using Eq. (11).
2:   Inner loop: Maximize the right side of Eq. (12) with fixed $\boldsymbol{\theta}_{c_k}^{\Delta;t}$ and compute $\boldsymbol{\tau}_{c_k}^{t+1}(\mathbf{x}_{c_k})$ using the belief propagation algorithm proposed in Liu and Ihler (2012):

$$m_{k \to l}(\mathbf{z}_{c_k}) \propto \psi_{s_{kl}}(\mathbf{z}_{s_{kl}}) \sum_{\mathbf{z}_{c_k \setminus s_{kl}}} \frac{\sigma\left[\psi_{c_k}(\mathbf{z}_{c_k}) m_{\sim k}(\mathbf{z}_{c_k})\right]}{m_{l \to k}(\mathbf{z}_{c_k})},$$

where $\quad \psi_{c_k}(\mathbf{z}_{c_k}) = \exp[\boldsymbol{\theta}_{c_k}^{\Delta}(\mathbf{z}_{c_k})], \quad$ and $\quad \sigma[\boldsymbol{\tau}_{c_k}(\mathbf{z}_{c_k})] = \begin{cases} \boldsymbol{\tau}_{c_k}(\mathbf{z}_{c_k}) & c_k \in \mathcal{R} \\ \boldsymbol{\tau}_{c_k}(\mathbf{z}_{c_k}) \boldsymbol{\tau}_{c_k}(\mathbf{d}_{c_k}|\mathbf{x}_{c_k}) & c_k \in \mathcal{A}, \end{cases}$

with $\quad \boldsymbol{\tau}_{c_k}(\mathbf{z}_{c_k}) = \psi_{c_k}(\mathbf{z}_{c_k}) m_{\sim k}(\mathbf{z}_{c_k})$ and $\quad \boldsymbol{\tau}_{c_k}(\mathbf{x}_{c_k}) = \max_{\mathbf{d}_{c_k}} \sum_{\mathbf{y}_{c_k}, a} \boldsymbol{\tau}_{c_k}(\mathbf{z}_{c_k})$

**Output:** The local policies $\{\boldsymbol{\tau}(d_i|\mathbf{x}(\Gamma_i))\}$, and the value function $\hat{v}(\mathbf{x}) = \exp(\Phi)\boldsymbol{\tau}(\mathbf{x})$.

---

## 6 Experiments

We perform experiments in two domains, disease management in crop fields and viral marketing, to evaluate the performance of our factored variational value iteration algorithm (FVI). For comparison, we use approximate policy iteration algorithm (API) (Sabbadin et al., 2012), (a mean-field based policy iteration approach), and the approximate linear programming algorithm (ALP) (Guestrin et al., 2001). To evaluate each algorithm's performance, we obtain its approximate local policy, then compute the expected value of the policy using either exact evaluation (if feasible) or a sample-based estimate (if not). We then compare the expected reward $U^{alg}(\mathbf{x}) = \frac{1}{|\mathcal{X}|} \sum_{\mathbf{x}} v^{alg}(\mathbf{x})$ of each algorithm's policy.

### 6.1 Disease Management in Crop Fields

A graph-based MDP for disease management in crop fields was introduced in (Sabbadin et al., 2012). Suppose we have a set of crop fields in an agricultural area, where each field is susceptible to contamination by a pathogen. When a field is contaminated, it can infect its neighbors and the yield will decrease. However, if a field is left fallow, it has a probability (denoted by $q$) of recovering from infection. The decisions of each year include two options ($D_i = \{1, 2\}$) for each field: cultivate normally ($d_i = 1$) or leave fallow ($d_i = 2$). The problem is then to choose the optimal stationary policy to maximize the expected discounted yield. The topology of the fields is represented by an undirected graph, where each node represents one crop field. An edge is drawn between two nodes if the fields share a common border (and can thus pass an infection). Each crop field can be in one of three states: $x_i = 1$ if it is uninfected and $x_i = 2$ to $x_i = 3$ for increasing degrees of infection. The probability that a field moves from state $x_i$ to state $x_i + 1$ with $d_i = 1$ is set to be $P = P(\varepsilon, p, n_i) = \varepsilon + (1 - \varepsilon)(1 - (1 - p)^{n_i})$, where $\varepsilon$ and $p$ are parameters and $n_i$ is the number of the neighbors of $i$ that are infected. The transition function is summarized in Table 1. The reward function depends on each field's state and local decision. The maximal yield $r > 1$ is achieved by an uninfected, cultivated field; otherwise, the yield decreases linearly with the level of infection, from maximal reward $r$ to minimal reward $1 + r/10$. A field left fallow produces reward 1.

Table 1: Local transition probabilities $p\left(x_i'|x_{N(i)}, a_i\right)$, for the disease management problem.

|  | $d_i = 1$ | | | $d_i = 2$ | | |
|---|---|---|---|---|---|---|
|  | $x_i = 1$ | $x_i = 2$ | $x_i = 3$ | $x_i = 1$ | $x_i = 2$ | $x_i = 3$ |
| $x_i' = 1$ | $1 - P$ | 0 | 0 | 1 | $q$ | $q/2$ |
| $x_i' = 2$ | $P$ | $1 - P$ | 0 | 0 | $1 - q$ | $q/2$ |
| $x_i' = 3$ | 0 | $P$ | 1 | 0 | 0 | $1 - q$ |

### 6.2 Viral Marketing

Viral marketing (Nath and Domingos, 2010, Richardson and Domingos, 2002) uses the natural premise that members of a social network influence each other's purchasing decisions or comments; then, the goal is to select the best set of people to target for marketing such that overall profit is

maximized. Viral marketing has been previously framed as a one-shot influence diagram problem (Nath and Domingos, 2010). Here, we frame the viral marketing task as an MDP planning problem, where we optimize the stationary policy to maximize long-term reward.

The topology of the social network is represented by a directed graph, capturing directional social influence. We assume there are three states for each person in the social network: $x_i = 1$ if $i$ is making positive comments, $x_i = 2$ if not commenting, and $x_i = 3$ for negative comments. There is a binary decision corresponding to each person $i$: market to this person ($d_i = 1$) or not ($d_i = 2$). We also define a local reward function: if a person gives good comments when $d_i = 2$, then the reward is $r$; otherwise, the reward is less, decreasing linearly to minimum value $1 + r/10$. For marketed individuals ($d_i = 1$), the reward is 1. The local transition $p\left(x_i'|x_{N(i)}, d_i\right)$ is set as in Table 1.

### 6.3 Experimental Results

We evaluate both problems above on two topologies of model, each of three sizes (6, 10, and 20 nodes). Our first topology type are random, regular graphs with three neighbors per node. Our second are "chain-like" graphs, in which we order the nodes, then connect each node at random to four of its six nearest nodes in the ordering. This ensures that the resulting graph has low tree-width ($\leq 6$), enabling comparison of the ALP algorithm. We set parameters $r = 10$ and $\varepsilon = 0.1$, and test the results on different choices of $p$ and $q$.

Tables 2– 4 show the expected rewards found by each algorithm for several settings. The best performance (highest rewards) are labeled in bold. For models with 6 nodes, we also compute the expected reward under the optimal *global* policy $\pi^*(\mathbf{x})$ for comparison. Note that this value over-estimates the best possible local policy $\{\pi_i^*(\Gamma_i(\mathbf{x}))\}$ being sought by the algorithms; the best local policy is usually much more difficult to compute due to imperfect recall. Since the complexity of the approximate linear programming (ALP) algorithm is exponential in the treewidth of graph defined by the neighborhoods $\Gamma_i$, we were unable to compute results for models beyond treewidth 6.

The tables show that our factored variational value iteration (FVI) algorithm gives policies with higher expected rewards than those of approximate policy iteration (API) on the majority of models ($156/196$), over all sets of models and different $p$ and $q$. Compared to approximate linear programming, in addition to being far more scalable, our algorithm performed comparably, giving better policies on just over half of the models ($53/96$) that ALP could be run on. However, when we restrict to low-treewidth "chain" models, we find that the ALP algorithm appears to perform better on larger models; it outperforms our FVI algorithm on only $4/32$ models of size 6, but this increases to $14/32$ at size 10, and $25/32$ at size 20. It may be that ALP better takes advantage of the structure of $\mathbf{x}$ on these cases, and more careful choice of the cluster graph could similarly improve FVI.

The average results across all settings are shown in Table 5, along with the relative improvements of our factored variational value iteration algorithm to approximate policy iteration and approximate linear programming. Table 5 shows that our FVI algorithm, compared to approximate policy iteration, gives the best policies on regular models across sizes, and gives better policies than those of the approximate linear programming on chain-like models with small size (6 and 10 nodes). Although on average the approximate linear programming algorithm may provide better policies for "chain" models with large size, its exponential number of constraints makes it infeasible for general large-scale graph-based MDPs.

## 7 Conclusions

In this paper, we have proposed a variational planning framework for Markov decision processes. We used this framework to develop a factored variational value iteration algorithm that exploits the structure of the graph-based MDP to give efficient and accurate approximations, scales easily to large systems, and produces better policies than existing approaches. Potential future directions include studying methods for the choice of cluster graphs, and improved solutions for the dual approximation (12), such as developing single-loop message passing algorithms to directly optimize (12).

### Acknowledgments

This work was supported in part by National Science Foundation grants IIS-1065618 and IIS-1254071, a Microsoft Research Fellowship, National Natural Science Foundation of China (#61071131 and #61271388), Beijing Natural Science Foundation (#4122040), Research Project of Tsinghua University (#2012Z01011), and Doctoral Fund of the Ministry of Education of China (#20120002110036).

Table 2: The expected rewards of different algorithms on regular models with 6 nodes.

| (p,q) | Disease Management | | | | | Viral Marketing | | | |
|---|---|---|---|---|---|---|---|---|---|
| | Exact | FVI | API | ALP | | Exact | FVI | API | ALP |
| (0.2, 0.2) | 202.4 | **202.4** | 164.7 | 148.3 | | 259.3 | **258.2** | 250.0 | 237.7 |
| (0.4, 0.2) | 169.2 | **169.2** | 139.0 | 123.3 | | 212.2 | **195.3** | 192.6 | 183.4 |
| (0.6, 0.2) | 158.1 | 155.2 | **157.4** | 115.4 | | 209.6 | 167.8 | **174.0** | 156.4 |
| (0.8, 0.2) | 154.1 | 152.7 | **153.2** | 106.0 | | 209.5 | 152.7 | **172.2** | 144.7 |
| (0.2, 0.4) | 262.5 | **259.2** | 254.7 | 236.7 | | 361.6 | **361.6** | 355.8 | 355.0 |
| (0.4, 0.4) | 220.1 | **219.1** | 177.0 | 181.3 | | 300.2 | **285.8** | 285.1 | 267.3 |
| (0.6, 0.4) | 212.1 | **203.8** | 203.8 | 162.7 | | 297.3 | 244.6 | **249.6** | 244.8 |
| (0.8, 0.4) | 211.7 | **198.2** | 198.2 | 136.1 | | 297.3 | 225.2 | **296.8** | 273.5 |
| (0.2, 0.6) | 349.3 | **349.3** | 333.6 | 307.3 | | 428.1 | **428.1** | 428.1 | 427.7 |
| (0.4, 0.6) | 290.7 | **276.7** | 276.7 | 200.0 | | 361.8 | **351.7** | 303.3 | 350.0 |
| (0.6, 0.6) | 284.7 | 242.7 | **243.7** | 212.8 | | 355.5 | 304.7 | 152.5 | **306.5** |
| (0.8, 0.6) | 284.0 | **236.1** | 236.1 | 194.7 | | 355.5 | 282.9 | **355.0** | 271.3 |
| (0.2, 0.8) | 423.6 | **423.6** | 423.6 | 274.7 | | 470.0 | **469.8** | 469.8 | 469.8 |
| (0.4, 0.8) | 362.2 | **351.0** | 344.3 | 264.5 | | 411.6 | 402.0 | 402.0 | **403.7** |
| (0.6, 0.8) | 351.6 | **304.8** | 302.7 | 242.5 | | 398.2 | 347.8 | **351.8** | 336.6 |
| (0.8, 0.8) | 350.5 | **284.2** | 284.9 | 207.9 | | 398.0 | 320.8 | **398.0** | 294.0 |

Table 3: The expected rewards of different algorithms on "chain-like" models with 10 nodes.

| (p,q) | Disease Management | | | Viral Marketing | | |
|---|---|---|---|---|---|---|
| | FVI | API | ALP | FVI | API | ALP |
| (0.3, 0.3) | **304.8** | 258.4 | 288.9 | **355.5** | 324.1 | 335.5 |
| (0.5, 0.3) | 273.4 | 228.7 | **292.7** | 308.1 | 291.5 | **323.8** |
| (0.7, 0.3) | 262.2 | 261.6 | **329.6** | **298.5** | 298.1 | 269.7 |
| (0.3, 0.5) | 420.2 | 395.4 | **456.5** | **550.1** | 523.9 | 543.9 |
| (0.5, 0.5) | **358.5** | 317.7 | 302.6 | **453.3** | 450.9 | 410.0 |
| (0.7, 0.5) | 343.8 | 344.9 | **394.3** | 386.1 | 418.6 | **436.9** |
| (0.3, 0.7) | 612.9 | **613.6** | 531.2 | 659.9 | 634.8 | **664.7** |
| (0.5, 0.7) | 498.2 | 491.8 | **538.6** | **542.7** | 523.9 | 518.2 |
| (0.7, 0.7) | **430.0** | 411.8 | 427.3 | **496.9** | 495.7 | 451.2 |

Table 4: The expected rewards ($\times 10^2$) of different algorithms on models with 20 nodes.

| (p,q) | Disease Manag. | | Viral Marketing | | | (p,q) | Disease Manag. | | Viral Marketing | |
|---|---|---|---|---|---|---|---|---|---|---|
| | FVI | API | FVI | API | | | FVI | API | FVI | API |
| (0.2, 0.2) | **7.17** | 6.33 | 7.87 | **7.88** | | (0.4, 0.2) | **5.93** | 5.19 | **6.53** | 5.65 |
| (0.6, 0.2) | **5.33** | 4.94 | **5.99** | 5.28 | | (0.8, 0.2) | 5.12 | **5.20** | **5.76** | 5.62 |
| (0.4, 0.4) | **9.10** | 8.82 | **11.56** | 11.52 | | (0.4, 0.4) | **7.70** | 6.23 | **9.23** | 8.83 |
| (0.4, 0.4) | **7.04** | 6.17 | **7.95** | 7.65 | | (0.4, 0.4) | **6.72** | 6.72 | **7.45** | 7.14 |
| (0.6, 0.6) | **12.29** | 12.11 | **13.85** | 13.85 | | (0.6, 0.6) | 9.97 | **10.06** | **11.74** | 11.72 |
| (0.6, 0.6) | 8.50 | **8.72** | **10.22** | 10.02 | | (0.6, 0.6) | **8.01** | 7.69 | **9.23** | 8.88 |
| (0.8, 0.8) | 14.53 | **14.57** | 15.25 | **15.27** | | (0.8, 0.8) | **12.57** | 12.43 | **13.47** | 13.22 |
| (0.8, 0.8) | **10.90** | 10.78 | **11.82** | 11.50 | | (0.8, 0.8) | **9.92** | 9.56 | **10.77** | 10.64 |

Table 5: Comparison of average expected rewards on regular and "chain-like" models.

| Type | $n = 6$ | | $n = 10$ | | $n = 20$ | |
|---|---|---|---|---|---|---|
| Regular | FVI: **275.8** API: 271.4 | Rel. Imprv. 1.6% | FVI: **458.7** API: 452.3 | Rel. Imprv. 1.4% | FVI: **935.6** API: 905.1 | Rel. Imprv. 3.37% |
| Chain | FVI: **275.8** API: 271.6 ALP: 244.9 | Rel. Imprv. 1.6% 12.6% | FVI: **415.7** API: 399.4 ALP: 414.7 | Rel. Imprv. 4.1% 0.7% | FVI: 821.9 API: 749.6 ALP: **872.2** | Rel. Imprv. 9.7% −5.8% |

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
