[Reviews · NeurIPS 2013]

Submitted by Assigned_Reviewer_2

This work presents a variational planning framework for finite and infinite MDPs. A finite MDP can be viewed as an influence diagram, and thus all steps from [13] are directly followed: solving using the backpropagation algorithm [2] which is an optimization problem at each step, finding the dual problem which resembles the free-energy functional optimization, and applying the Bethe relaxation for efficiently getting an approximate solution using belief propagation.
The experimental part shows that this method typically brings higher rewards for datasets with problems in two domains: disease management in crop fields and viral marketing.

This work appears to me as a direct application of [13] from influence diagrams specifically to MDPs. The authors mention themselves that MDPs could be viewed as influence diagrams. Indeed, the differences between the works seem mainly in notation. The only part I'm not sure about is how novel is the extension to infinite MDPs.
Summary: This work directly applies previous results [13] from influence diagrams to MDPs, while also mentioning that MDPs (or at least finite MDPs) can be viewed as influence diagrams. Novelty is not clear.

Submitted by Assigned_Reviewer_3

Variational Planning for Factored MDPs

This papers studies the problem of approximately solving factored MDPs. It takes the mixed loopy BP algorithm of Liu and Ihler [13] as a starting point. The algorithm in [13] would work on finite horizon MDPs, since these can be rolled out to form a specific influence diagram. For infinite horizon MDPs the algorithm of [13] doesn't work out of the box. This paper proposes a double loop algorithm that uses the algorithm of [13] in the inner loop as an approximations that leverages the factored nature of the problem. The double loop algorithm is motivated by Theorem 2: "For an infinite MDP, we can simply consider a two-stage MDP with the variational form in Eq. (5) but with this additional constraint [that v^{t-1}(x)=v^t(y) when x=y]"

Quality : An extension of the proof of 2 would be required. It is not immediately clear that the dual form for a general influence diagram can be directly applied to the fixed point equations of an infinite horizon MDP. Here the proof is restating the claim.

The experiments compare to well-known algorithms for solving general MDPs. Why are there no algorithms from the factored MDP literature?

Clarity : the paper is very hard to read. I had to re-read it twice to have a reasonable guess what the paper tried to achieve. For example the title of section 4 is ambiguous: with "infinite MDPs" are there variables that can take on an infinite number of values or is the horizon infinite? If the latter is intended, the standard jargon is "infinite horizon MDP".
If the key trick in the paper is the observation above theorem 2 and theorem 2 itself, this can be easily explained in an abstract.

Originality : to the best of my knowledge the use of the algorithm of [13] as a building block to solve infinite horizon mdps is new.

Significance : better approximations for (factored) MDPs are important for many applications.
Summary: A promising trick to use the loopy BP algorithm from Liu and Ihler for influence diagrams to factored infinite horizon MDPs. The paper is hard to read, proofs could be extended, and the experiments do not compare to any algorithms for factored MDPs.

Submitted by Assigned_Reviewer_5

This paper introduces a variational framework for planning in infinite-horizon factored MDPs. Leveraging previous work by Liu and Ihler [13], a variational “dual” representation of the maximum expected reward problem (Bellman equation) is considered. Exploiting the factored structure of the transition probabilities and the additive form of the rewards, they introduce an approximation which can be solved by a double-loop Belief-propagation style algorithm.
This algorithm is shown to outperform approximate policy iteration and approximate linear programming on several instances of a disease management and a viral marketing MDP.

The main contribution of this paper is an extension of the previous approach of Liu and Ihler [13] from influence diagrams to an infinite horizon MDP case. The contribution is mostly incremental with respect to [13], as the theoretical analysis mostly adapts and leverages the results in [13], and the proposed algorithmic technique is also very similar to the double loop algorithms in [13]. That being said, the adaptation and formalization is not entirely trivial so I think the paper is still interesting, and would be a very useful reference for practitioners who want to apply variational planning to MDPs. I also liked the rather extensive experimental evaluation, providing good experimental evidence that the technique actually works in practice. Although it's not groundbreaking, I think there is value in this paper, both for people working on MDPs and for the community interested in variational inference techniques, as it provides a new application domain.

The paper is very well written. It is quite dense with some heavy math, but it is generally well explained and the authors usually provide an intuitive explanation. One issue is that the paper is not entirely self contained. I found it difficult to understand the details without reading [13] first. I understand space limitations, but it would be good to provide more background if possible.
A more detailed step-by-step algorithmic description of algorithm 1 would be also be useful. For example, I couldn't find a definition of v_{ck,x}(y_ck,a) in eq (11). How does one compute v_{ck,x}(y_ck,a) in Eq. (11) in terms of the inputs of algorithm 1 on line 1 of the pseudocode?
The not entirely standard notation (like ~ to indicate product of messages as in [13]) in the pseudocode should also be described.

I wish more details were given on the cluster graph part. On line 238-242 it seems that there is a constructive procedure. In the pseudocode, it is given as input. How is one supposed to construct the cluster graph? How was that done for the experiments?

I like the extensive experimental comparison. I wish more details were given on the runtime / convergence of belief propagation. For reproducibility, it would be good to specify how the cluster graph is chosen, and how are the messages initialized.

In what sense is the optimization problem solved (line 267)? is it guaranteed to find a locally optimal solution? does the policy improve at each iteration of the double loop?

Is the value function v* output by algorithm 1 equal to the actual expected reward of the local policies (returned by the algorithm 1) or is it an approximation? Either way, it would be good to use a different notation from the one in theorem 2 (b) (if I understand correctly, they could be different because of the approximations introduced)

minor things:

186-188 is tau* the solution of the optimization problem?
204 I couldn't find a definition of Phi(Theta)
218 woulc -> “would”
233 it seems to me it should be v_\gamma(y,a) instead of v_\gamma(x,a)
258 the second time should probably be tau_ck(z_ck) instead of tau_ck(x_ck)
261 shouldn't the discount factor gamma appear somewhere in (11)?
266-268 the notation used for the entropies seems inconsistent (missing semicolon)
272, 275-276 what is the difference between the two different notations of tau_ck (with and without subscript x)?
287 what is Phi(Theta)? is it as in equation (12)?
reference [11] is incomplete
Summary: This paper introduces a nice, but not overly profound extension of a recent variational framework for structured decision-making (for influence diagrams) to the infinite-horizon factored MDP case. They adapt a previous belief-propagation style algorithm that leverages the factored structure of the problem, which is shown to perform well in practice on several MDP benchmarks.

Submitted by Assigned_Reviewer_7

SUMMARY:
This article proposes an adaptation of the approach of Liu and Ihler 2012 (ref [13]), to the approximate resolution of Graph-based MDPs (refs [10-12]), a particular form of Factored MDPs.
At first (Section 2), MDPs and Factored MDPs are reviewed. Note that what the authors call Factored MDPs are in fact Graph-based MDPs. This is different, and I don't think the proposed approach can be readily extended to Factored MDPs in general.
Then (Section 3), the authors show how to adapt the general framework of [13] (devoted to policy optimisation in influence diagrams) to the particular case of finite-horizon MDPs.
The real innovation of the paper is presented in Sections 4 and 5. In Section 4, the result concerning finite horizon MDPs is extended to infinite horizon MDPs (the proof being rather straightforward). Then, Section 5 proposes a variational algorithm for Graph-based MDPs. This algorithm is tested on benchmarks prensented in [10-12] and compared to the existing ALP and API algorithms. The result are, globally, in favor of the original algorithm.

QUALITY :
The paper is interesting, presents a nice theoretical result, together with a consistent experimental study. The misleading claim that the paper proposes an approach for planning in (general) Factored MDPs should be given up and restricted to planning in GMDPs instead (and Section 2.2 should be renamed accordingly). Apart from that,the paper is globally well-written (just avoid to call the algorithm "Backwards Induction" "Backwards Reduction"!)...

CLARITY :
The paper is globally clear. My only concerns about clarity are in identifying the original contribution of the paper (see below).

ORIGINALITY :
This is (maybe with significance, see below) my main concern about the paper. It is difficult, in its current form, to identify the original contribution of the paper.
- Up to Section 3, this is only state of the art.
- I guess Section 4 is original (is it?), but trivial.
- In Section 5, I guess that the additive-multiplicative reward is not original (already in [13]). By the way, I was wondering how would the transformation behave in the case where f is an addition of N factors: does a take N values, or 2^N?
- In Section 5, results (8) to (12) are new, but look very like rather straightforward extensions of [13]. However, I am not sure of that and would be happy to know what precisely is difficult in going from the results in [13] to the results here...

SIGNIFICANCE:
To my opinion, the paper is and is not significant...
- It is significant since it takes an existing problem (GMDP), provides a new algorithm (derived in a not completely straightforward way from an existing algo) which shows better results than existing approaches on small problems. Furthermore, the variational approach is interesting and deserve to be disseminated.
- It is not that much significant in that (i) the approach is a derivative from [13] (even if the derivation is not straightforward), (ii) the considered problem (GMDP) is not as general as the authors implicitly claim (FMDP) and (iii) the experiments are only made on small problems (20 nodes), when state of the art approaches (ALP/API) solve problems with several hundreds of nodes.
Concerning point (iii), I am surprised, given the effort that has been made to implement existing approaches, that larger problems have not been tested and that no results about computation time have been given. This omission leaves the feeling that the variational approach is time inefficient and cannot be applied to large-scale problems. Even if this is true, it is worse for the paper to omit this point than mentionning it, in my opinion...

Summary: Plus: The variational approach of [13] is then extended to the GMDP problem and shows nice results on some problems.
Minus: The paper's presentation is misleading and seems to exagerate the contribution. GMDPs are solved and not FMDPs, the novelty of the approach compared to [13] and [10-12] is not easy to grasp, experiments are also misleading, leaving the feeling that the approach outperforms existing approaches, while neither tackling large problems, nor precising computation times...
Author Feedback

Author rebuttal: We thank all the reviewers for their helpful comments and suggestions. We will use them to improve the final version, including adding detailed proofs, correcting typos, providing additional background and more details on cluster graphs, clarifying “finite and infinite horizon MDPs”, and so on. Here are some issues that we think were the most important to respond.

1. Although our proposed algorithm is based on the results of Liu & Ihler 12, we argue that our work has significant impact and novelty. (a) Note that our main contribution is in solving *infinite* (not finite) horizon graph-based MDPs, which is an important but extremely difficult problem for which very few efficient algorithms exist so far. We demonstrate that our algorithm outperforms the existing methods on this task. More importantly, our framework connects this problem to the variational inference literature, and could open the door for many additional algorithms or motivate more work in this direction. (b) Although seemingly straightforward in hindsight, the insight extending methods for finite horizon MDPs to infinite horizon MDPs is itself valuable and technically non-trivial.

2. R3, R5, and R7 mentioned that we compare our algorithm only with the approximate policy iteration algorithm and approximate linear programming algorithm on models with 20 nodes in the experimental section. The reasons are as follows. First, our algorithm can be easily applied to larger problems with hundreds of nodes and arbitrary structures; however, exact evaluation of the algorithms (computing the exact value functions given the obtained policies) becomes computationally infeasible on larger models. Second, the problems in our experiments do not have any specific contextual independence structure; this makes the structured value iteration and structured policy iteration algorithms extremely slow on these models and so we did not compare them in our results. Regarding runtime, our algorithm is comparable to the other algorithms we tested; we will add the run times in the final version.

3. We agree that “graph-based MDPs” is more precise than “factored MDPs” (in response to R7). We will change this term in the final version.